# Non-Contact Heart Rate Monitoring Method Based on Wi-Fi CSI Signal

**DOI:** 10.3390/s24072111

**Published:** 2024-03-26

**Authors:** Juncong Sun, Xin Bian, Mingqi Li

**Affiliations:** 1Shanghai Advanced Research Institute, Chinese Academy of Sciences, Shanghai 201210, China; sunjc1@shanghaitech.edu.cn (J.S.); bianx@sari.ac.cn (X.B.); 2School of Information Science and Technology, Shanghaitech University, Shanghai 201210, China

**Keywords:** Wi-Fi, Channel State Information (CSI), non-contact monitoring, heart rate, subcarrier selection

## Abstract

This paper introduces an innovative non-contact heart rate monitoring method based on Wi-Fi Channel State Information (CSI). This approach integrates both amplitude and phase information of the CSI signal through rotational projection, aiming to optimize the accuracy of heart rate estimation in home environments. We develop a frequency domain subcarrier selection algorithm based on Heartbeat to subcomponent ratio (HSR) and design a complete set of signal filtering and subcarrier selection processes to further enhance the accuracy of heart rate estimation. Heart rate estimation is conducted by combining the peak frequencies of multiple subcarriers. Extensive experimental validations demonstrate that our method exhibits exceptional performance under various environmental conditions. The experimental results show that our subcarrier selection method for heart rate estimation achieves an average accuracy of 96.8%, with a median error of only 0.8 bpm, representing an approximately 20% performance improvement over existing technologies.

## 1. Introduction

In recent years, as people’s concern for health has deepened, ubiquitous health monitoring has become a focal point for both researchers and the public. Traditional vital signs monitoring technologies mainly rely on wearable devices that directly contact the user’s body. While they ensure a certain level of accuracy in monitoring, they also bring inconveniences to users, such as discomfort from long-term wear and impracticality in daily use. To address these issues, researchers have begun exploring non-contact health monitoring solutions, especially non-contact vital signs monitoring technologies. These not only enhance user comfort, but also improve usability in practical monitoring environments, as they do not require continuous user involvement.

Some diseases, such as myocardial infarction [1], apnea [1], and sudden infant death syndrome [2], can be detected or prevented by monitoring abnormalities in vital signs. In many such diseases, the symptoms of patients may only appear in a short period of time, thus requiring long-term and continuous monitoring. However, due to limitations in medical resources and funding, long-term hospitalization observation is impractical for most people. Therefore, continuous and low-cost vital sign monitoring in the home environment is essential. The traditional vital sign monitoring scheme mainly relies on specialized sensors attached to the body, such as electrodes used for polysomnography (PSG) [3] and electrocardiogram (ECG) [4]. However, these devices are not suitable for use in home environments as they are expensive and may affect sleep quality. Other non-professional sensors based on attachments, such as pressure or acceleration sensors [5,6], also require contact with the body, which may cause inconvenience to users. Therefore, non-contact vital sign monitoring technology has received great attention, especially solutions based on vision and radio frequency (RF). However, lighting conditions limit computer vision-based solutions [7,8]. Meanwhile, devices used in traditional RF-based solutions [9,10], such as software defined radio or radar systems, are often costly and difficult to deploy.

In recent years, vital sign monitoring based on Wi-Fi has gained widespread attention due to its capability for non-contact sensing, relying on ubiquitously deployed and cost-effective Wi-Fi devices. The ability of Wi-Fi to detect vital signs arises from the fact that breathing and heartbeats cause deformations in the abdomen and chest, thereby affecting the propagation of Wi-Fi signals. Channel State Information (CSI) captures these changes [11], enabling the recovery of the required vital signs. CSI provides detailed, fine-grained physical information on how signals propagate from the transmitter to the receiver, including detailed amplitude and phase information for different subcarriers. However, since the trunk deformations caused by breathing and heartbeats are very subtle and have a relatively minor impact on Wi-Fi signal propagation, theoretical models are necessary to guide this sensing process. Currently, most advanced schemes are based on the Fresnel zone model [12], the Fresnel diffraction model [13], or the CSI-ratio model [14,15].

Existing research has primarily focused on estimating breathing rates [6], but in practical applications, human heart rate information is equally important, as many heart-related diseases cannot be prevented by monitoring breathing alone. Additionally, the phase information in CSI also contains breath and heart rate information, whereas most existing works only consider estimating from either amplitude or phase, potentially not fully utilizing the perceptive capabilities of the CSI signal.

To overcome these challenges, this paper proposes a non-contact heart rate monitoring method based on Wi-Fi CSI signals, which accurately estimates heart rate in a home environment by combining the amplitude and phase information of the CSI signal. Our contributions can be summarized as follows:

1. In this work, our method not only considers both the amplitude and phase information of the CSI signal but also combines them using a rotational projection method to enhance perception performance.

2. We propose a subcarrier selection algorithm based on frequency domain metrics and design a combined signal filtering and subcarrier selection algorithm to improve the accuracy of heart rate estimation.

3. Through extensive experimental testing, we validate the performance of our method under different conditions and demonstrate that our approach offers higher accuracy and a broader detection range compared to existing methods.

The remainder of this paper is organized as follows: Section 2 introduces related work, Section 3 details the proposed method, Section 4 presents the results and analysis of experimental testing, and finally Section 5 summarizes the contributions of this paper and discusses future work directions.

## 2. Related Work

The widespread availability of Wi-Fi devices and the convenience of wireless sensing technology have fostered the flourishing development of passive sensing research based on Wi-Fi [16,17,18]. Such studies typically focus on the Received Signal Strength Indicator (RSSI) or Channel State Information (CSI). Compared to CSI, RSSI is easier to obtain, but it offers relatively coarser granularity in sensing. Meanwhile, although acquiring CSI requires modifications to the low-level drivers of Wi-Fi cards, it provides finer sensing granularity compared to RSSI [19,20].

By leveraging Wi-Fi’s RSSI or CSI data, researchers have successfully realized various application scenarios using commercial off-the-shelf Wi-Fi devices (COTS Wi-Fi), including human presence detection [21], gesture recognition [22,23,24], cross-domain gesture recognition [25,26], localization [27], sleep motion detection [28], and driving activity detection [29]. Recently, the field of Wi-Fi sensing research has further expanded. For instance, Wang et al. [30] captured human keypoint signals using Wi-Fi devices, achieving human visualization without the need for visual equipment. Additionally, Wang et al. [31] demonstrated how to steal smartphone passwords using commercial Wi-Fi devices. Some studies have also attempted to optimize Wi-Fi sensing performance through advanced signal processing techniques. For example, the PhaseAnti project [19] effectively identified various activities under the same channel interference conditions by extracting signal components unrelated to channel interference. The SMARS project [32] combined CSI with Autocorrelation Function (ACF) technology, providing a new solution for detecting sleep respiratory status and sleep stages. These studies not only showcase the wide applications of Wi-Fi sensing technology in daily life and work, but also open new possibilities for the future development of wireless sensing technology.

In the existing field of vital signs monitoring [1,11], the approaches can generally be categorized into two types: contact-based and non-contact-based. Traditional contact-based methods typically rely on devices that directly contact the human body, such as wristbands and fingertip pulse oximeters, to assess an individual’s heart rate and breathing. However, these methods can be uncomfortable for users, especially when sensors need to be worn for extended periods.

In contrast, non-contact methods offer more comfortable and convenient solutions. For example, in [33,34], the authors discuss a non-invasive respiratory monitoring technology that combines Software Defined Radio (SDR) with machine learning algorithms for detecting COVID-19-related breathing patterns; some systems use the built-in microphone of smartphones [35] to capture sound information for estimating breathing rates, while others employ detectors connected to beds to estimate breathing and heart rates [16,17]. However, these methods have some limitations, such as short sensing distances, sensitivity to environmental noise, and limited applicability. Additionally, camera-based systems have been proposed in [18,19], where videos or images capturing chest movements during sleep are used to estimate breathing and heart rates. Although these systems can achieve high accuracy, they may raise privacy concerns and are susceptible to low-light conditions.

As an alternative, research on sensing using radio frequency (RF) signals has been gaining increasing attention recently. When RF signals propagate from a transmitter to a receiver, they are influenced by chest and abdominal movements caused by breathing and heartbeats. However, solutions based on RF signals typically rely on specialized equipment, such as ultra-wideband devices [35] or Frequency-Modulated Continuous Wave (FMCW) radars [9,36]. The devices used in these solutions are expensive and not suitable for everyday environments. In contrast, Wi-Fi-based solutions are more cost effective, simpler to deploy, and can be implemented using commercial off-the-shelf (COTS) devices.

In recent studies, methods for heartbeat detection using Wi-Fi Channel State Information (CSI) have made some progress. However, most research tends to choose either the amplitude or the phase of the CSI signal for detection [37,38,39,40]. This approach overlooks the complementarity of amplitude and phase in vital signs detection using CSI signals [41]. Although studies [42,43] have considered the complementarity of amplitude and phase, in practical solution design, they still only choose either the amplitude or the phase signal that offers superior sensing performance. This selective approach may not fully utilize the informational content of CSI signals, thereby limiting further improvements in the accuracy of heartbeat detection.

In our research, we adopted a comprehensive approach that thoroughly considers both the amplitude and phase information of the CSI signal. By introducing the technique of rotational projection, we organically combined amplitude and phase information and successfully extended this projection method to the field of heartbeat detection. Further, we innovatively proposed a new Wi-Fi CSI subcarrier selection algorithm, which utilizes frequency domain metrics to select subcarriers. Moreover, we constructed a heart rate estimation framework based on the ‘subcarrier heart rate interval fusion method’. This framework, by integrating information from different subcarriers, provides a more effective method for heart rate estimation.

## 3. Proposed Method

In the introduction to the related work, several advantages of the presented research were highlighted in comparison to other solutions. First, both the amplitude and phase information of Wi-Fi CSI signals are comprehensively utilized by the proposed method rather than relying solely on amplitude or phase difference information, which is a significant deviation from previous studies. In past approaches [37,38,39,40], typically, only the signal components with superior sensing performance were chosen for analysis. The rotational projection method, previously used in breath detection [41], has been adapted and extended to heartbeat detection. Additionally, a new frequency domain metric, the Heartbeat to Subcomponent Ratio (HSR), has been proposed for effectively evaluating the performance of subcarriers in heartbeat detection. Based on HSR, a novel subcarrier selection method has been introduced. This method involves first identifying the maximum frequency energy intervals of each subcarrier, then determining the most common shared maximum energy frequency interval among these, which is assumed to be the interval closest to the actual heart rate. Consequently, the top five subcarriers with the highest HSR values are selected, the maximum frequency peaks are found using FFT, and the information from these selected subcarriers is integrated for heart rate estimation. The innovations of the method compared to other approaches can be seen in the following Table 1.

In this research, the heart rate detection scheme is composed of three core parts: data preprocessing, subcarrier selection, and heart rate estimation, as shown in Figure 1.

During the data preprocessing phase, four key steps are designed: downsampling, computing the CSI ratio, Savitzky–Golay (SG) filtering, and rotational projection for preliminary signal processing. Downsampling is aimed at reducing computational complexity. The computation of the CSI ratio ensures the simultaneous use of both amplitude and phase difference information of the CSI signal. SG filtering is utilized to detect and eliminate outliers in the original CSI ratio data, thereby smoothing the signal. The rotational projection, inspired by previous work in breath detection, combines the real and imaginary parts of the CSI signal to integrate both amplitude and phase difference information. After rotational projection, a candidate set of subcarriers for selection is obtained.

In the subcarrier selection phase, the preprocessed set of subcarriers is filtered for subsequent analysis. The signal is then decomposed, and frequency domain metrics are applied to select subcarriers based on the decomposed signal. Specifically, the preprocessed CSI data are filtered using the Heartbeat to Subcomponent Ratio (HSR) metric, followed by Discrete Wavelet Transform (DWT) decomposition to further extract heartbeat information. Subsequently, subcarriers are selected based on the Common Maximum Energy (CME) frequency interval. Initially, the energy ratio for each subcarrier in different frequency domain windows is calculated, recording the window with the highest energy ratio. Next, the most common frequency interval across all subcarriers is determined, suggesting that the heartbeat frequency lies within this range. Lastly, the top five subcarriers with the highest HSR values within this frequency interval are calculated. A Fourier Transform (FFT) is performed on these subcarrier signals, and their peak frequencies are calculated. To consolidate the information from these five subcarriers, their normalized HSR values are used as weighting coefficients to calculate the weighted average frequency peak. This frequency peak is then multiplied by a time factor to obtain the final heart rate estimation.

The subsequent sections elaborate on the details of each of these components.

### 3.1. Data Preprocessing

For the proposed preprocessing of the original Channel State Information (CSI) data, the goal is to enhance data quality through a series of refined processing steps and to lay the groundwork for subsequent analysis. The following steps are involved in preprocessing.

#### 3.1.1. Data Downsampling

As shown in To reduce the volume of data and lessen the computational burden in subsequent processing steps while ensuring that key information is retained, downsampling techniques are employed on the high-sampling-rate collected CSI data. Specifically, the original data sampling frequency is reduced to 30 Hz.This approach effectively eliminates data redundancy and enhances computational efficiency.

#### 3.1.2. CSI Ratio

In line with previous work [14], the CSI ratio model is introduced to enhance signal sensing capability. First, the concept of CSI is briefly introduced. In complex indoor environments, radio frequency signals reach the receiver via multipath propagation, which includes direct paths and various reflected paths. CSI represents the aggregate of these multipath signals. Mathematically, CSI can be expressed [41] as follows:(1)$$H\left(f,t\right)=\sum_{i=1}^{L}A_{i}e^{-j2\pi \frac{d_{i}\left(t\right)}{\lambda }}.$$

In the formula, *L* represents the number of paths, $A_{i}$ is the complex gain, and $d_{i}\left(t\right)$ is the propagation length of the *i*th path. Based on previous work, CSI can be divided into static and dynamic components. The static component includes signals from the direct path and other fixed reflection paths, while the dynamic component corresponds to variations caused by human target movement, as shown in Figure 2. Therefore, CSI can be rewritten [41] as follows:(2)$$H\left(f,t\right)=H_{s}\left(f,t\right)+H_{d}\left(f,t\right)=H_{s}\left(f,t\right)+A\left(f,t\right)e^{-j2\pi \frac{d(t)}{\lambda }}.$$

In the equation, $H_{s}\left(f,t\right)$ represents the static component, while A(f,t), $e^{-j2\pi \frac{d(t)}{\lambda }}$, and d(t) correspond to the complex attenuation, phase shift, and path length, respectively, of dynamic component $H_{d}\left(f,t\right)$.

As shown, the time-varying CSI phase caused by changes in the dynamic component $H_{d}\left(f,t\right)$ is expressed as
(3)$$\theta =\beta -\alpha =\beta -\arcsin\frac{P}{\left|H\right|}=∠H_{s}-\arcsin\frac{|H_{d}|\sin\rho }{\sqrt{|H_{s}{|}^{2}+|H_{d}{|}^{2}+2|H_{s}\left|\right|H_{d}|\cos\rho }},$$
where β is the phase of static component $H_{s}$, α is the phase difference between the total CSI *H* and $H_{s}$, *P* is the distance from the edge of the dynamic component $H_{d}$ to $H_{s}$, |H| is the amplitude of the total CSI, and ρ is the Fresnel phase, essentially the phase difference between the dynamic and static components. Since $|H_{d}|\sin\rho$ is relatively small compared to $|H_{s}|$ [19], we approximate the CSI phase as follows:(4)$$\theta \approx ∠H_{s}-\frac{|H_{d}|}{\sqrt{|H_{s}{|}^{2}+|H_{d}{|}^{2}+2|H_{s}\left|\right|H_{d}|\cos\rho }}\sin\rho .$$

We note that $|H_{d}|$ is much smaller than $|H_{s}|$, and the cosine term cosρ in the denominator has almost no effect on the second term on the right side of Equation (Equation 3). Thus, the CSI phase is mainly determined by sinρ, and the waveform of the CSI phase is a quasi-sinusoidal waveform. The CSI amplitude can be expressed as
(5)$$\left|H\right|=\sqrt{|H_{s}{|}^{2}+|H_{d}{|}^{2}+2|H_{s}\left|\right|H_{d}|\cos\rho }.$$

It can be seen that it is a quasi-cosine waveform.

Mathematically, the complementarity of CSI amplitude and phase difference in the field of vital signs monitoring manifests as complementary sensing capabilities. Specifically, when the sensing capability of the CSI amplitude is strong, the sensing capability of the phase difference tends to be weak as shown in Figure 3, and vice versa. This phenomenon has been confirmed in previous studies. Based on this fact, it is believed that, as demonstrated in breath monitoring, a comprehensive and integrated consideration of both CSI amplitude and phase difference can provide more effective support for heartbeat detection.

The CSI ratio model greatly facilitates the exploration of the complementary characteristics of CSI. This model calculates the ratio of the CSI readings between two antennas at the receiver. Through this division operation, most of the pulse noise and transient noise in the signal can be effectively eliminated. The mathematical expression of the CSI ratio [14] is as follows:(6)$$\frac{H_{1}\left(f,t\right)}{H_{2}\left(f,t\right)}=\frac{e^{-j\theta_{\mathrm{offset}}}\left(H_{s,1}+A_{1}e^{-j2\pi \frac{d_{1}\left(t\right)}{\lambda }}\right)}{e^{-j\theta_{\mathrm{offset}}}\left(H_{s,2}+A_{2}e^{-j2\pi \frac{d_{2}\left(t\right)}{\lambda }}\right)}.$$

In this expression, $H_{s,1}$ and $H_{s,2}$ represent the static component parts of CSI readings from the first and second antennas, respectively. $A_{1}$ and $A_{2}$ represent the complex attenuation of the dynamic component in the CSI readings from the first and second antennas, respectively. $d_{1}\left(t\right)$ and $d_{2}\left(t\right)$ represent the lengths of different dynamic reflection paths, and $e^{-j\theta_{\mathrm{offset}}}$ represents the inherent time-varying phase offset of commercial Wi-Fi.

As shown in Figure 4a,b, after the original collected signal is down-sampled, by utilizing the CSI ratio model and performing division operations, not only is the main pulse noise and transient noise in the signal significantly reduced as shown in Figure 4c, but the model also integrates the information of CSI amplitude and phase difference, theoretically enhancing the perception efficiency of the CSI signal. For this reason, the CSI ratio model is adopted in this study for heartbeat detection.

#### 3.1.3. SG Filtering

To further enhance signal quality, a Savitzky–Golay filter is employed in this study for smoothing the CSI ratio signal. This filter’s advantage lies in its ability to smooth data while preserving signal characteristics, achieved by fitting data points to a local polynomial and replacing the original data points with the polynomial’s calculated values. The efficacy of the Savitzky–Golay filter is derived from its mathematical expression, which employs convolution operations and polynomial fitting for data smoothing, effectively improving the signal-to-noise ratio. The formula for the SG filter can be expressed as
(7)$$y_{i}^{\prime }=\sum_{j=-\frac{m}{2}}^{\frac{m}{2}}c_{j}y_{i+j},$$
where $y_{i}^{\prime }$ is the filtered data point, *m* is the chosen window size, $c_{j}$ is the coefficients in the convolution kernel, and $y_{i+j}$ is the original data and its neighbors. In practical applications, the $c_{j}$ coefficients are pre-calculated based on the chosen window size and polynomial degree.

The SG filter optimizes the local approximation performance of the data, accurately depicting the overall trend of the data even for individual data points affected by noise. As shown in Figure 4d, therefore, the SG filter significantly improves the quality of signal smoothing while maintaining data structure.

#### 3.1.4. Rotational Projection

In this study, the first task undertaken was to extend the projection method used in breath detection in [14] to heartbeat detection. First, this method is briefly introduced.

When performing CSI measurements on a single antenna, the phase information itself is not suitable for monitoring due to time-varying random phase offsets. However, by using the ratio of CSI, stable phase difference information between two antennas can be obtained (since the random phase offsets are canceled out). This allows for the integration of the phase and amplitude information of the CSI ratio to overcome blind spots in monitoring, thus extending the range of perception.

As shown in Figure 5, it is important to note that a complex number can be represented in the form of a+bi and $Ae^{i\theta }$, where *a* and *b* represent the real part (I) and the imaginary part (Q), respectively, while *A* and θ represent the amplitude and phase. The real part (I component) and the imaginary part (Q component) of the CSI ratio exhibit perfect complementarity, meaning that if the I component at a certain location is not suitable for heartbeat monitoring, the Q component might be advantageous, and vice versa. Therefore, these two components are combined, effectively unifying amplitude and phase for monitoring. By assigning different weights to the composite I and Q components, a series of candidate combinations of subcarrier signals can be generated, and the designed subcarrier selection method is used to filter out the subcarrier signals most suitable for heartbeat monitoring for further analysis.

Unlike the traditional method of choosing the better one from the I and Q components, we linearly combine the I and Q components by projecting them onto the complex plane along different angles θ. The figure shows how to project z=a+bi onto the axis [cosθ,sinθ] to obtain a new point $z^{\prime }$, where θ is the angle of the projection axis. According to simple geometry, we can derive
(8)$$oz^{\prime }=\left[cos\theta ,sin\theta \right]\left[\begin{array}{c} a\\ b \end{array}\right]=acos\theta +bsin\theta .$$

Specifically, point *z* is a linear combination of its I and Q components, where the weight of the I component is cosθ and the weight of the Q component is sinθ. Similarly, for the time series of the CSI ratio data *x*, its projection *y* onto the axis [cosθ,sinθ] can be represented as
(9)$$y_{e}=\left[cos\theta ,sin\theta \right]\left[\begin{array}{c} R(x)\\ I(x) \end{array}\right],$$
where R(x) is the real part (I component) of *x*, and I(x) is the imaginary part (Q component) of *x*. By varying θ within the range of 0 to 2π in fixed steps, we can generate a variety of different candidate combinations.

### 3.2. Subcarrier Selection

#### 3.2.1. Composite Signal Selection

In the above-described CSI data preprocessing steps, a candidate set of signals for each subcarrier was generated, which includes signals obtained by combining the real and imaginary parts of the CSI using the rotational projection method. The goal is to select the best signal for each subcarrier after rotational projection, which is achieved through the defined metric, Heartbeat to Subcomponent Ratio (HSR). Next, the definition of HSR and its application in heartbeat detection is elaborated.

HSR (Heartbeat to Subcomponent Ratio) is defined as the ratio of the maximum FFT energy bin to the second largest FFT energy bin within the heartbeat frequency range. This ratio directly measures the characteristics of the energy distribution of the heartbeat signal:(10)$$\mathrm{HSR}=\frac{H_{E\_first}}{H_{E\_second}},$$
where $H_{E\_first}$ represents the largest FFT energy bin within the signal’s heartbeat range, and $H_{E\_second}$ represents the second largest FFT energy bin within this range. As shown in Figure 6, we propose that the higher the HSR value of a signal, the stronger its ability to perceive heartbeat motion.

Unlike common subcarrier selection metrics such as variance, spectral stability score, and HNR (Harmonic-to-Noise Ratio), HSR evaluates heartbeat characteristics based on changes in energy distribution, overcoming the limitation of variance methods that focus only on time-domain waveform changes and ignore spectral features. HSR does not rely on spectral historical consistency and can acutely capture the dynamic changes of heartbeat signals, thereby outperforming the spectral stability score method. Compared to HNR, HSR avoids sensitivity to same-frequency interference when assessing the quality of heartbeat signals, enhancing the stability and accuracy of heartbeat detection. In summary, HSR quantifies the heartbeat energy characteristics directly and efficiently by analyzing the ratio of the main energy bins.

The HSR (Heartbeat to Subcomponent Ratio) metric is utilized to filter and select from the candidate combinations for each subcarrier. As can be seen from the aforementioned illustration, the HSR values of the combined signals, obtained after rotational projection, increase, indicating an enhanced ability to perceive heartbeats as shown in Figure 7. The signals with the highest HSR values are ultimately selected for subsequent operations. This selection process ensures that the most effective signals for heartbeat detection are being used, thereby increasing the accuracy and reliability of the monitoring system.

Since the time-domain waveform of the heartbeat is more difficult to discern clearly compared to breathing, the focus is primarily on observing its frequency domain changes. The rotational projection on the complex plane effectively amplifies the components of the signal that are most relevant to the heartbeat while reducing the influence of other irrelevant or noise components. This process highlights the heartbeat signal in the frequency domain, making it more distinct and easier to analyze. Consequently, this approach significantly aids in the accurate detection and estimation of heart rates, especially in scenarios where the heartbeat signal may be weak or obscured by noise.

#### 3.2.2. Discrete Wavelet Transform

The Discrete Wavelet Transform (DWT) is a method for analyzing signals at various resolutions. It decomposes a signal into approximations and details at different scales by applying a series of wavelet filters and downsampling operations, achieving multi-scale analysis of the signal. Therefore, DWT is applied to the signals selected from the candidate set to extract heartbeat signals.

First, the fourth wavelet of the Daubechies series, commonly abbreviated as ‘db4’, is chosen. This wavelet is often used for processing signals with prominent features because it balances the need for time and frequency localization. Daubechies wavelets, known for their good compact support properties, perform well with signals displaying sharp transitions.

The choice of the number of decomposition levels is dependent on the signal’s sampling rate and the frequency range of interest. In this study, with the signal sampling rate at 30 Hz, the aim is to cover a heartbeat frequency range of approximately 0.8 Hz to 2.5 Hz. Considering that each level of decomposition roughly halves the frequency range, choosing 4 levels of decomposition allows for the analysis of signals close to $30/2^{4}=1.875\;\mathrm{Hz}$, encompassing the typical frequency range of heartbeat signals.

Finally, wavelet decomposition is performed on the selected subcarrier signals to extract heartbeat information. This step is crucial as it filters out irrelevant frequencies and focuses on the range where heartbeat information is present, thereby enhancing the signal’s relevance to the task of heartbeat detection.

As shown in Figure 8, it can be observed that the combined signal undergoes four levels of wavelet decomposition. The signal at the fourth level closely resembles the true waveform of the heart rate. This level of the signal, the closest approximation to the actual heart rate waveform, is selected as the heartbeat signal *H* for subsequent processing. This selection is based on the principle that higher levels of wavelet decomposition provide a more detailed and refined view of the lower frequency components of the signal, crucial for heart rate detection. The fourth level of decomposition isolates the frequency components corresponding to the heart rate, effectively filtering out higher frequency noise and other irrelevant signal components. Using this decomposed signal at the appropriate level ensures that the subsequent analysis and heart rate estimation are based on the most relevant and accurate representation of the heart rate signal, thereby improving the precision and reliability of the monitoring system.

#### 3.2.3. Subcarrier Selection

As shown in Figure 9, after applying Discrete Wavelet Transform (DWT), the reconstructed heartbeat signals *H* for each subcarrier are obtained. For each subcarrier signal *h*, a Fast Fourier Transform (FFT) is performed. A frequency domain sliding window, with parameters including window length *L* and step size *s*, is designed. This sliding window traverses all the FFT bins, calculating energy ratio $R_{i}$ for each window *w* against the entire signal, *h*. The maximum energy ratio $R_{i\_MAX}$ and its corresponding frequency window $w_{i}$ are identified. Repeating this process for each subcarrier signal yields a set of frequency bands $W=\{w_{1},w_{2},\ldots ,w_{n}\}$. The most common frequency band $w_{c}$ within this set is then found, which is the most likely heart rate interval.

Finally, the signals of the top five subcarriers with the highest HSR values within the common frequency band $w_{c}$ are selected, recording their HSR values $r_{1},r_{2},r_{3},r_{4},r_{5}$ and the highest peak frequencies $f_{1},f_{2},f_{3},f_{4},f_{5}$. These subcarriers, with the highest HSR values in the common band, are deemed most suitable for heart rate estimation. This process effectively identifies the subcarrier signals that are most sensitive and accurate for detecting the heart rate, providing a robust method for heart rate monitoring using Wi-Fi CSI signals.

### 3.3. Heart Rate Estimation

After the selection of the five subcarriers most suitable for detecting heartbeats, the final step is the estimation of the heart rate. The maximum frequency peaks of these five subcarriers are first found. Their HSR values are then normalized to define them as weight coefficients. The final heart rate estimate is calculated by multiplying these maximum frequency peaks with their normalized weight coefficients. Finally, this frequency estimate is multiplied by a time factor to obtain the heart rate estimate.

The formula for the heart rate estimate *E* is given by
(11)$$E=60\times \frac{r_{n1}\cdot f_{1}+r_{n2}\cdot f_{2}+r_{n3}\cdot f_{3}+r_{n4}\cdot f_{4}+r_{n5}\cdot f_{5}}{r_{n1}+r_{n2}+r_{n3}+r_{n4}+r_{n5}},$$
where *E* represents the final heart rate estimate, $r_{ni}$ the normalized HSR weight value of the selected subcarrier, and $f_{i}$ the maximum frequency peak of the subcarrier.

This method effectively combines the contributions of each selected subcarrier, weighted by their respective HSR values, to produce a more accurate and robust estimate of the heart rate. The multiplication by 60 converts the frequency estimate into beats per minute (BPM), which is the standard measure for heart rate.

## 4. Performance Evaluation

In this section, a comprehensive series of experiments focused on real-world application scenarios were conducted. The experimental configuration and environmental setup were detailed, describing the equipment configuration, the conditions during the experiments, and the relevant application parameters as shown in Table 2. Following this, the experimental results obtained using the selected data processing methods were presented, including a detailed analysis of their effectiveness in various scenarios.

The evaluation then moved on to a critical assessment of the performance of the method proposed in the study. This involved analyzing the accuracy, reliability, and efficiency of the method, comparing it with existing methods and benchmarks in the field. Finally, additional factors that might influence the system’s performance, such as environmental variables, hardware limitations, and other external influences, were explored.

### 4.1. Experimental Setup

As shown in Figure 10, the experiments in this study were conducted in two real-world scenarios, as illustrated in the following setup. The first experimental environment, simulating a home living room, was set up in a conference room furnished with tables, chairs, and cabinets, with participants seated in chairs for the experiment as shown in Figure 10a. The second experimental setting, simulating a home bedroom, was situated in a school dormitory, equipped with a bed and a table, where subjects conducted experiments while on the bed as shown in Figure 10b.

For the experiments, a wireless router was used as the signal transmitter and a laptop equipped with an Intel 5300 wireless card served as the receiver. In terms of data collection, data were gathered from nine participants over a month at a frequency of 200 Hz. To obtain the true heart rate values, data were also recorded using a Lepu heart rate monitor. Moreover, to consider the impact of different transmitter and receiver placement scenarios on the results, the line of sight (LOS) distance between the user and the devices was adjusted. Finally, the collected data were processed and analyzed using Matlab.

### 4.2. Data Processing Results

#### 4.2.1. Data Preprocessing Module

The data preprocessing module in this study primarily addressed the challenges of noise reduction, signal smoothing, and the integrated use of amplitude and phase information from the original CSI data. The original CSI signal, illustrated in Figure 11a, exhibited a high sampling rate and a wide amplitude range, indicating the presence of various signal components and noise. As shown in Figure 11b, downsampling reduced the complexity for subsequent computations.

Further refinement was achieved by calculating the CSI ratio from multiple antennas, which helped to stabilize phase information and is shown in Figure 12. Following that, the real and imaginary parts of the CSI ratio were combined, enhancing the perception of the heartbeat motion and resulting in a clearer pattern indicative of a rhythmic component corresponding to the heartbeat, as shown in Figure 13.

The signal was then subjected to multi-level wavelet decomposition, isolating the frequency components associated with the heart rate from other unrelated frequencies, as depicted in Figure 14. The fourth level of this decomposition produced a signal closely resembling the expected heartbeat waveform. The frequency spectrum of this decomposed signal revealed distinct peaks within the heart rate frequency band.

#### 4.2.2. Subcarrier Selection Module

Utilizing the proposed method, the target subcarriers were identified using their frequency domain peaks to accurately estimate the heart rate, illustrated in Figure 15a,b.

As shown in Figure 15b, we then used Equation (Equation 11) to calculate the weighted maximum peak frequency of the five selected subcarriers as the estimated heart rate frequency value.

### 4.3. Heart Rate Estimation Performance Evaluation

In this study, the Cumulative Distribution Function (CDF) was utilized to assess the estimation error, which is the deviation between the measured values and the true values. The CDF graph presents the cumulative probability distribution of the estimation errors. As benchmarks for comparison, methods based on variance, Harmonic-to-Noise Ratio (HNR), and spectral stability scoring were selected. Comparative analysis clearly demonstrated that the proposed method outperformed these three methods in terms of performance as shown in Figure 16.

Specifically, with the implemented method, 80% of the test errors were less than two beats per minute (bpm), and 90% of the test cases had errors less than 4.1 bpm. Moreover, the median error was reduced from 1 bpm to approximately 0.8 bpm, achieving an about 20% performance improvement compared to the previously best performing system. Additionally, within a detection distance range of 2 to 3 m, the method also showed superiority over existing methods. These results not only prove the effectiveness of the method in improving detection accuracy, but also demonstrate its potential in extending the detection range.

### 4.4. Impact of Other Factors

#### 4.4.1. Impact of Experimental Environment

The graph demonstrates the impact of different experimental environments on heart rate estimation when the distance between the transmitter and receiver is fixed at two meters. It was observed that the accuracy of heart rate detection remained above 96.5% in both the conference room and dormitory environments. This finding underscores the environmental adaptability of the method, indicating that it can maintain a high level of detection accuracy across various settings as shown in Figure 17.

#### 4.4.2. Impact of Transceiver Distance

During the experiments, tests were conducted at distances of 1 m, 2 m, 3 m, and 4 m between the transmitter and receiver. The charts created illustrate the influence of the distance between the transmitter and the receiver in the laboratory on the accuracy of heart rate estimation. The results showed that the highest accuracy of heart rate detection was achieved at a distance of 1 m, with a median error in detection of less than 1 bpm. However, as the distance between the transmitter and receiver increased, the accuracy of detection declined. Notably, at a distance of 4 m, the precision of detection dropped to its lowest as shown in Figure 18. This phenomenon is primarily due to the fact that at greater distances, the heartbeat-induced changes in the reflected signal are weaker, thus affecting detection precision.

#### 4.4.3. Impact of Distance from Person to Transceiver LOS

In the experiments, the effect of the line-of-sight (LOS) distance between the user and the transmitter–receiver setup was tested, including distances of 0.5 m, 1 m, 1.5 m, 2 m, 2.5 m, and 3 m. The experimental results indicated that within a range of 2 m, the detection error was very small, approximately around 1 bpm. However, as the distance increased, the accuracy of detection began to decline. Particularly at the distance of 3 m, the detection error increased to about 2.5 bpm as shown in Figure 19. These results highlight the impact of distance on the precision of heartbeat detection.

## 5. Conclusions

This paper investigates the problem of non-contact heart rate monitoring using Wi-Fi Channel State Information (CSI) and presents a comprehensive heart rate detection scheme that includes data preprocessing, subcarrier selection, and heart rate estimation steps. Utilizing the newly defined metric, Heartbeat to Subcomponent Ratio (HSR), we developed an innovative subcarrier selection method for heart rate estimation. Empirical results demonstrate that our proposed method for subcarrier selection in heart rate estimation achieves an overall accuracy rate above 96.5%, with a median error of only 0.8 bpm, marking an improvement of about 20% over existing solutions. In future research, we plan to expand upon the current method to further enhance the accuracy and range of detection and to extract more health-related information from heart rate data such as indicators of cardiac health and exercise rehabilitation metrics.

## Figures and Tables

**Figure 1 sensors-24-02111-f001:**
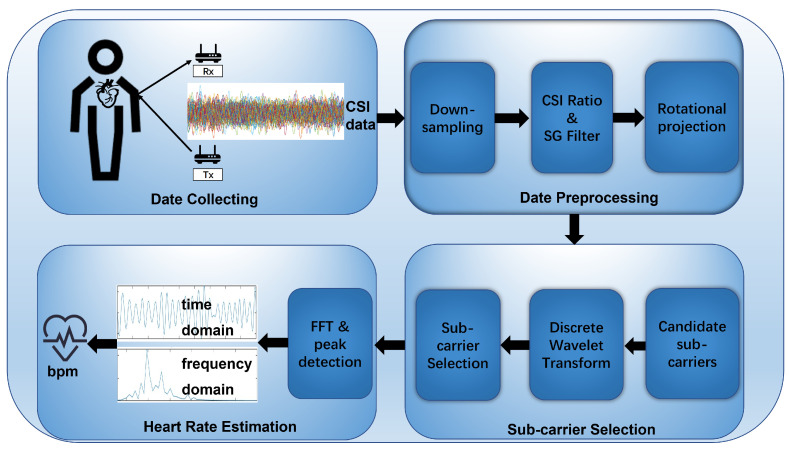
System overview.

**Figure 2 sensors-24-02111-f002:**
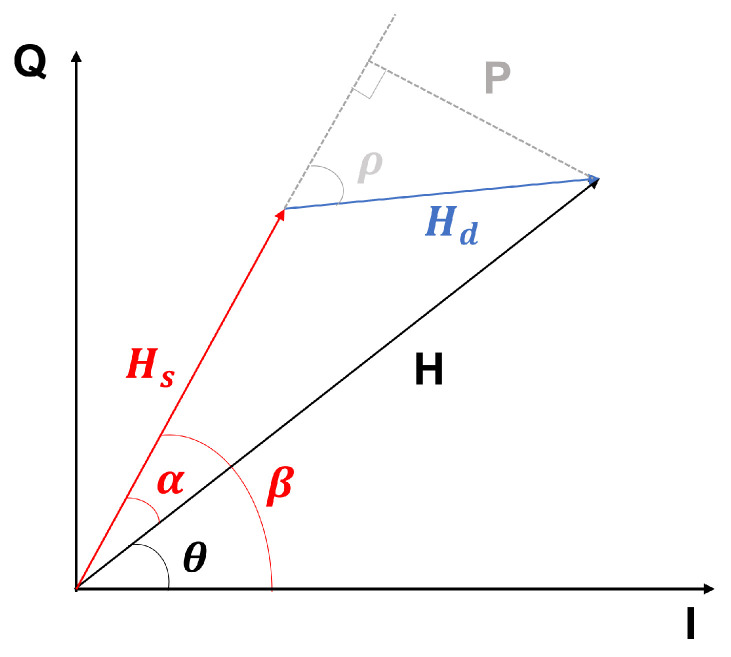
Changes in the dynamic and static components of CSI.

**Figure 3 sensors-24-02111-f003:**
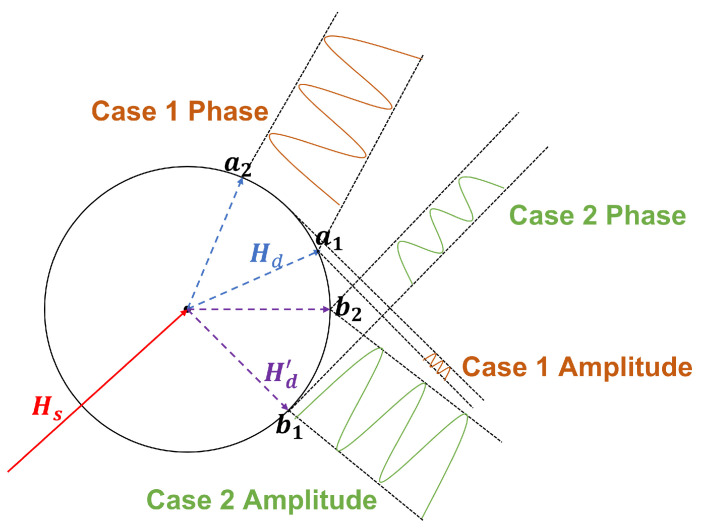
Complementarity of CSI amplitude and phase.

**Figure 4 sensors-24-02111-f004:**
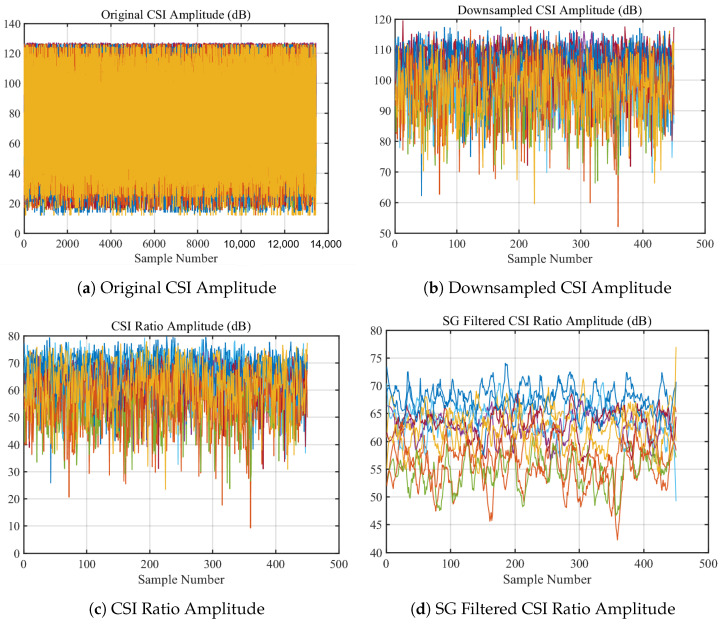
CSI raw signal preprocessing.

**Figure 5 sensors-24-02111-f005:**
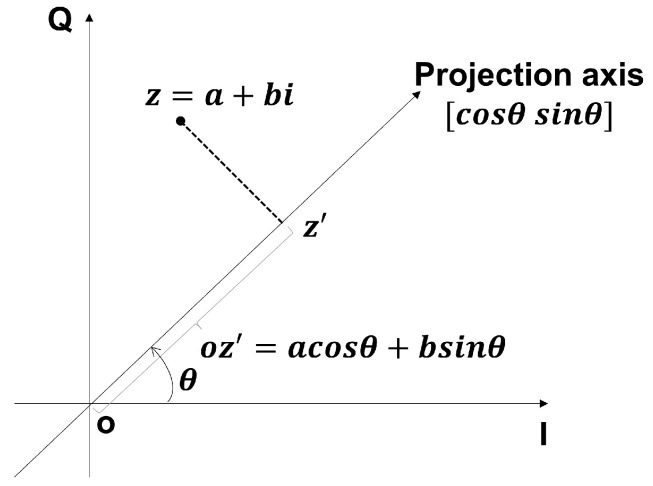
Complex plane rotation projection of CSI signals.

**Figure 6 sensors-24-02111-f006:**
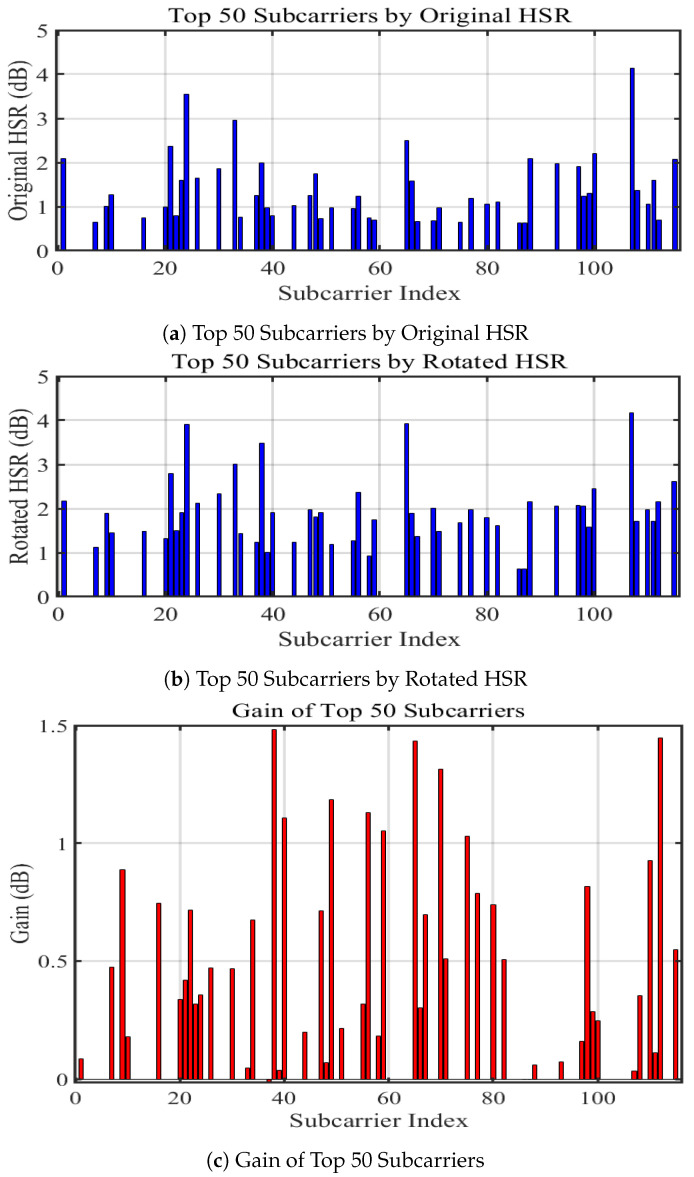
Changes in HSR values after rotational projection.

**Figure 7 sensors-24-02111-f007:**
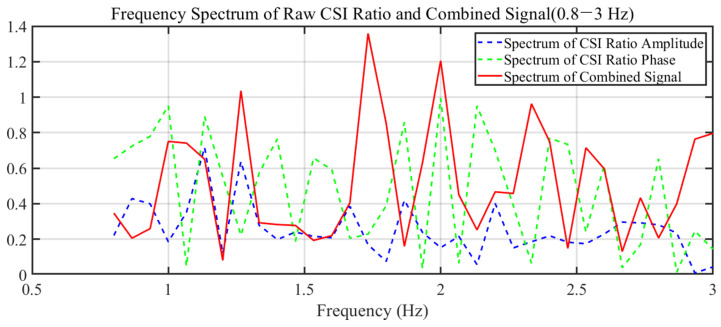
Frequency Spectrum of Raw CSI Ratio and Combined Signal.

**Figure 8 sensors-24-02111-f008:**
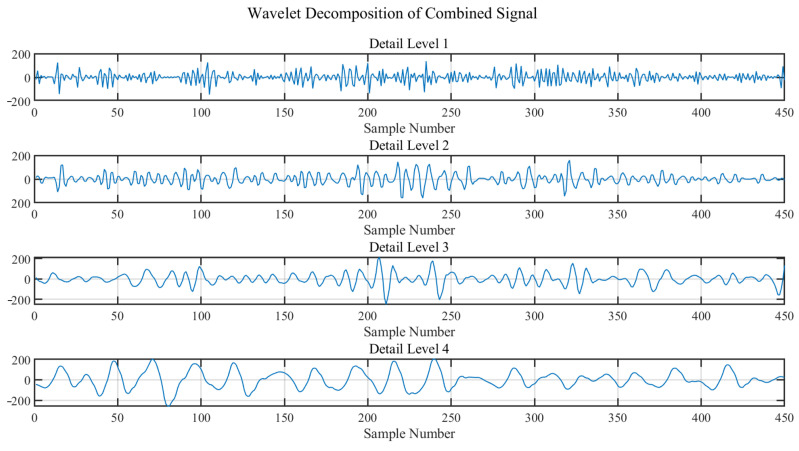
Wavelet Decomposition of Combined Signal.

**Figure 9 sensors-24-02111-f009:**
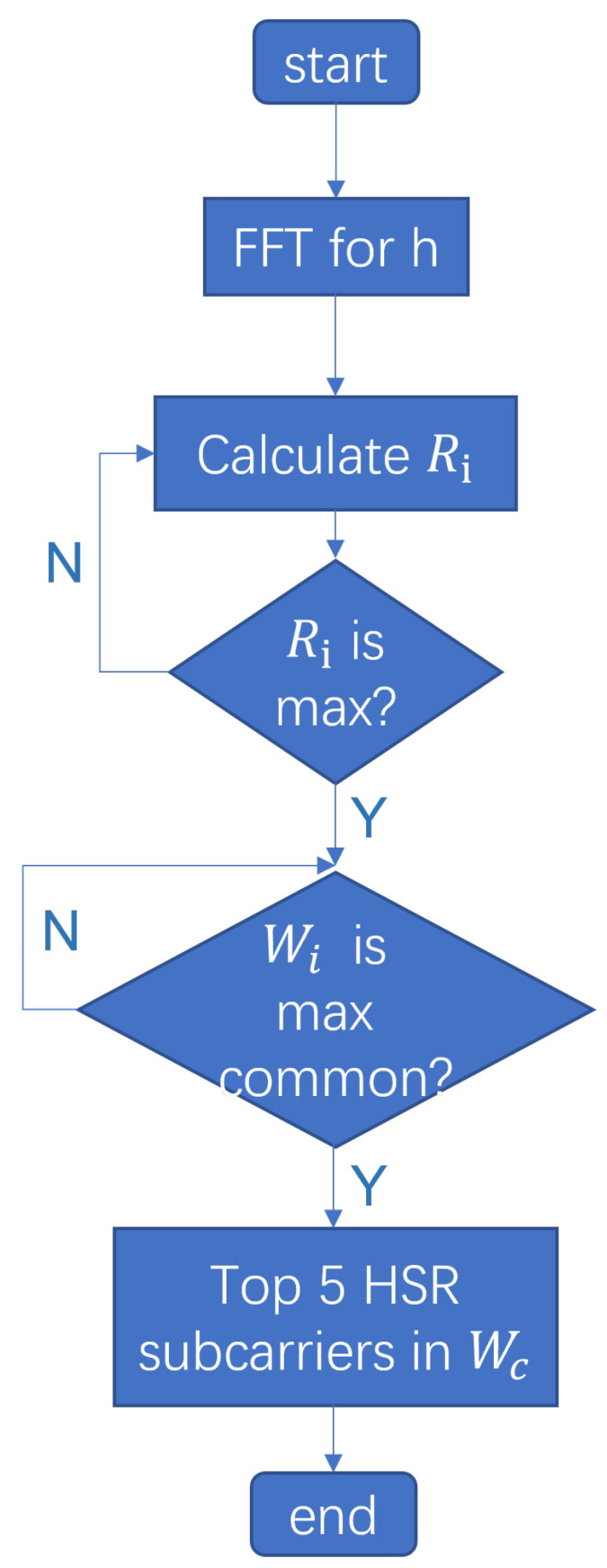
Flow chart of subcarrier selection algorithm.

**Figure 10 sensors-24-02111-f010:**
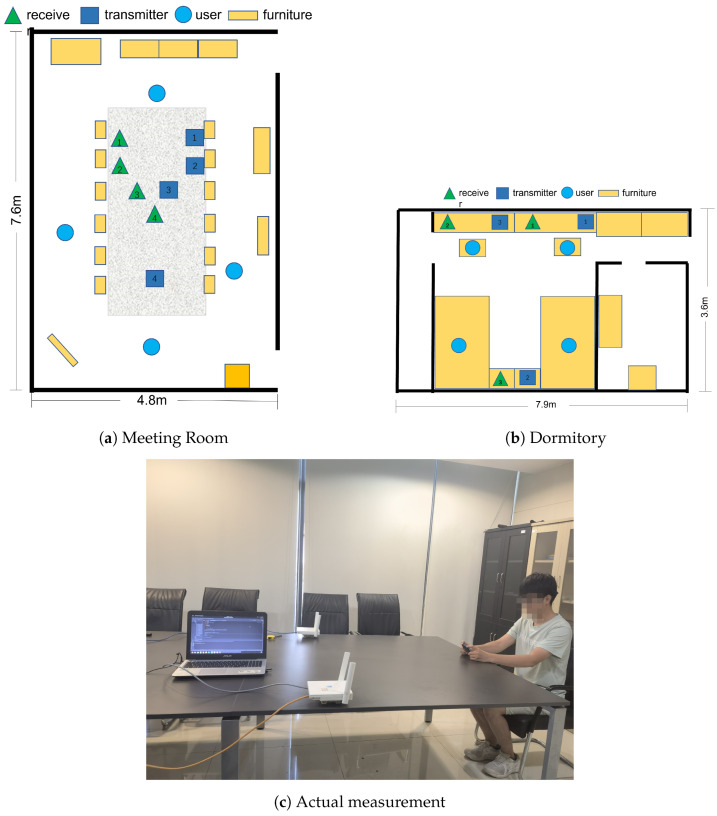
Experiment environments.

**Figure 11 sensors-24-02111-f011:**
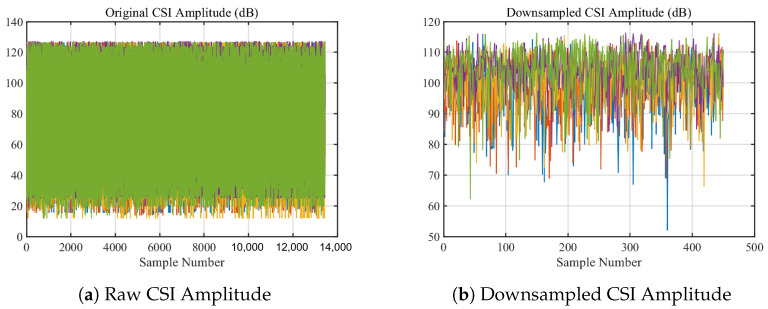
Data Preprocessing.

**Figure 12 sensors-24-02111-f012:**
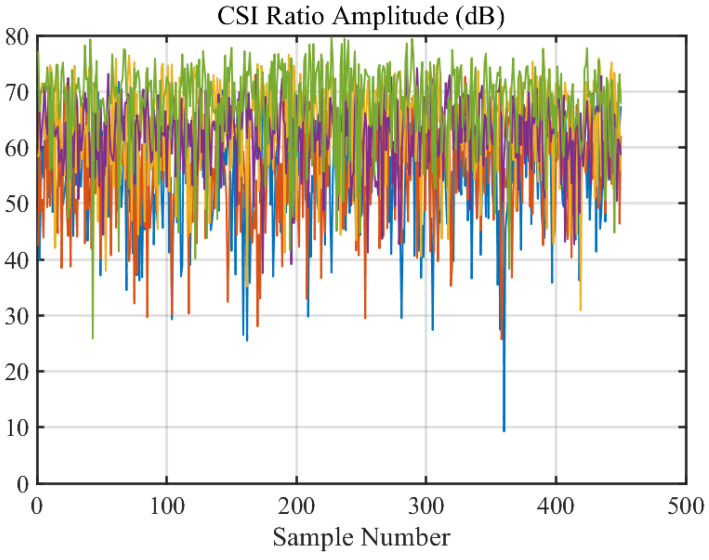
CSI Ratio.

**Figure 13 sensors-24-02111-f013:**
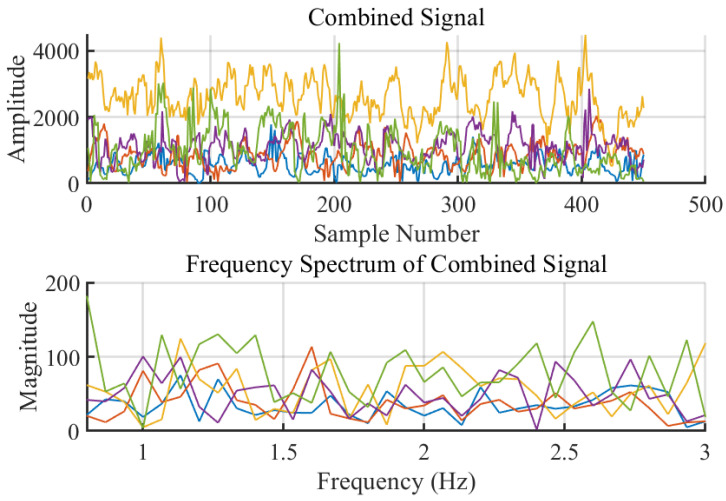
Combined Signal.

**Figure 14 sensors-24-02111-f014:**
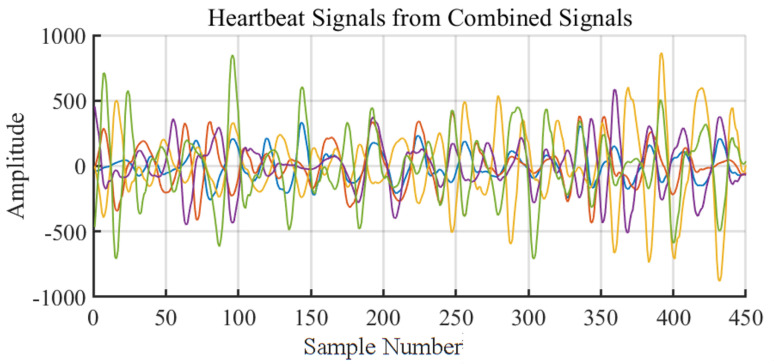
Heartbeat Signals from Combined Signals.

**Figure 15 sensors-24-02111-f015:**
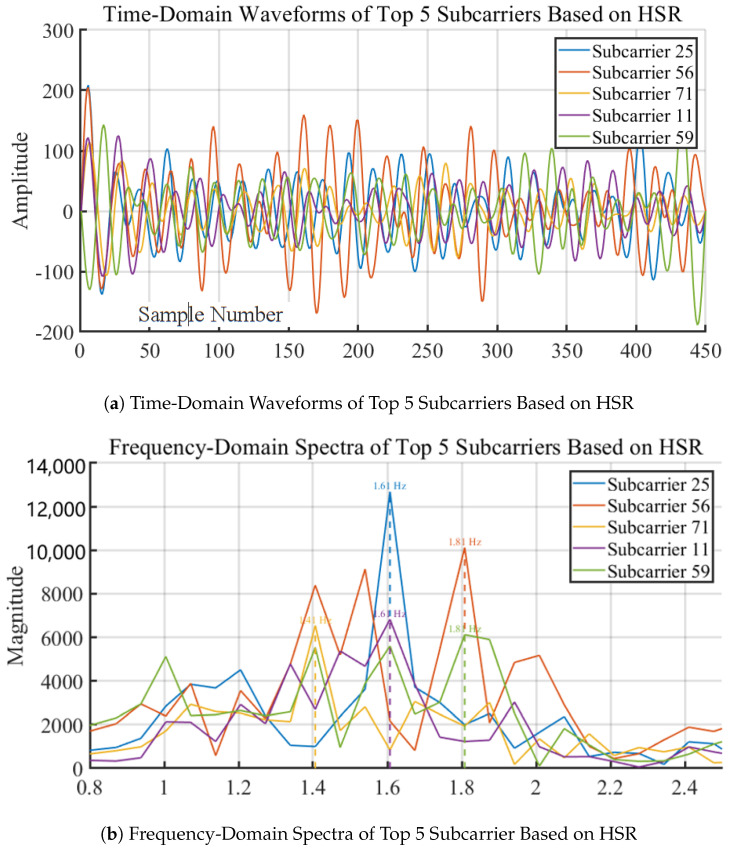
Subcarriers Selection.

**Figure 16 sensors-24-02111-f016:**
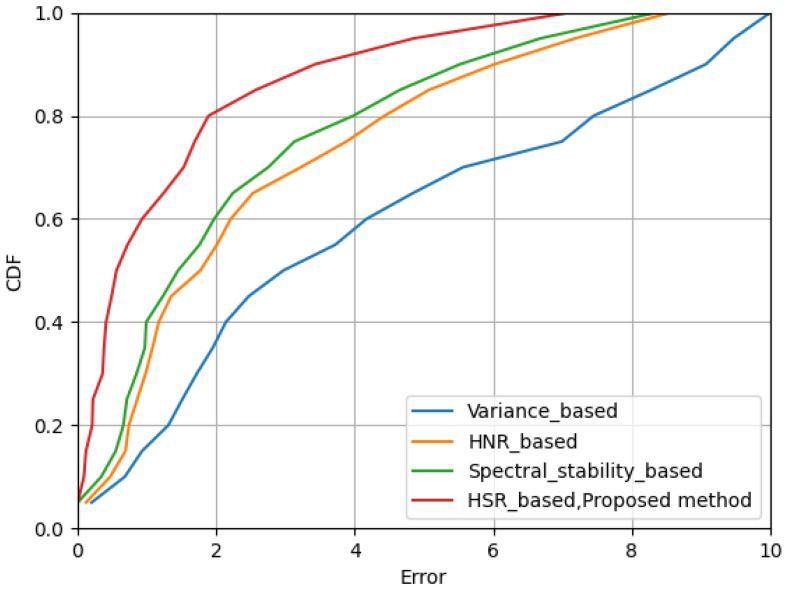
CDFs of estimation errors of heart rate estimation.

**Figure 17 sensors-24-02111-f017:**
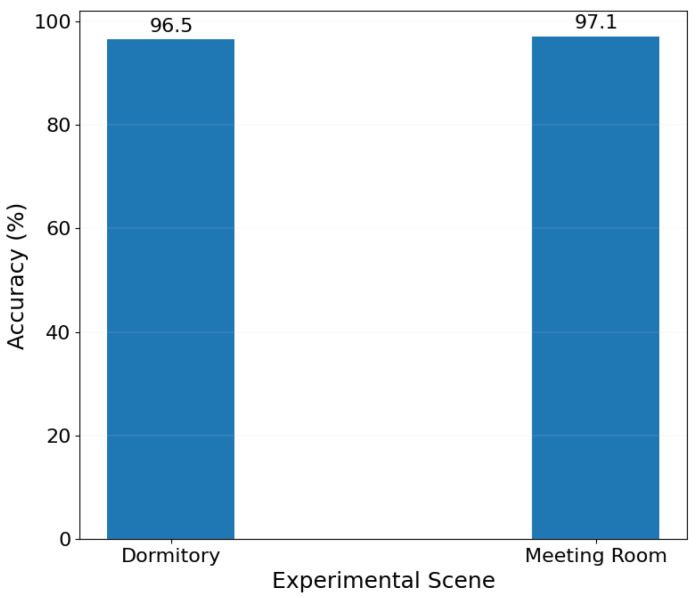
Imapct of the Experimental environment.

**Figure 18 sensors-24-02111-f018:**
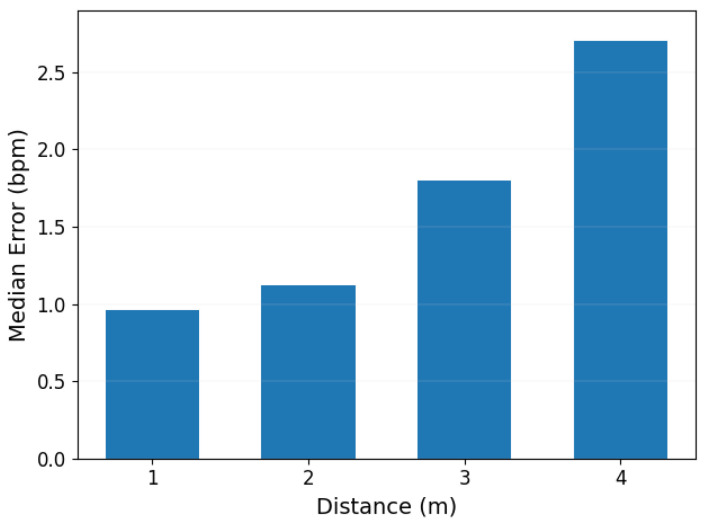
Impact of the Transmitter–Receiver Distance.

**Figure 19 sensors-24-02111-f019:**
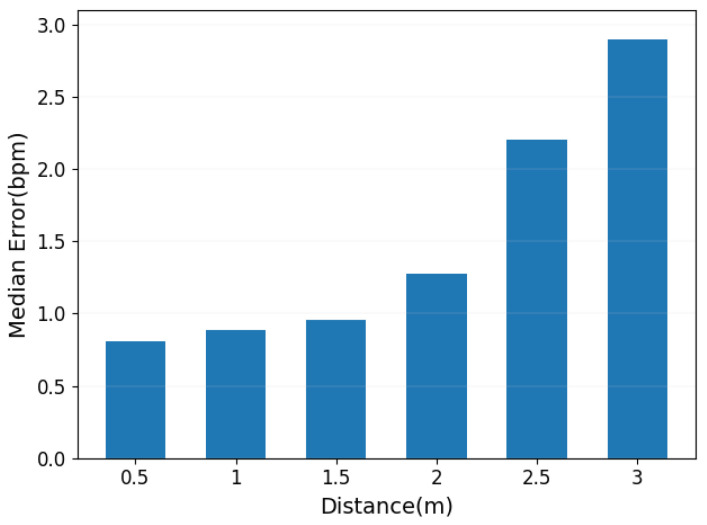
Impact of user-to-device distance.

**Table 1 sensors-24-02111-t001:** Comparison of Heart Rate Estimation Schemes.

Scheme	Selected Signal	Subcarrier Selection	Subcarrier	Model-Based	Detection Error (MedAE)	Estimate Method
[37]	Amplitude	Variance	Single	Y	2.28 bpm	FFT
[38]	Phase Difference	Mean Absolute Deviation	Single	Y	1.19 bpm	FFT
[39]	Phase Difference	Spectral Stability	Combination	Y	1.0 bpm	FFT
[40]	Phase Difference	Variance of Autocorrelation Function	Single	Y	1.0 bpm	FFT
[42]	Both	None	None	N	1.37 bpm	DL
[43]	Both, Selection	HNR	Combination	Y	1.39 bpm	FFT
Our Method	Both, Combination	HSR	Combination	Y	0.80 bpm	FFT

**Table 2 sensors-24-02111-t002:** Experimental Setup and Conditions.

Parameter	Setting
Experimental Location	Meeting room, Dormitory
Device Position	Transmitter and receiver 1 to 4 m apart, 1.5 m above the ground
Subject Position	0.5 to 3 m from the transmitter and receiver in Line of Sight (LOS)
Subjects	Nine individuals, six males and three females
Data Collection Equipment	Chuangtong Electronics TA8K (802.11 ac) custom model, ASUS laptop, Linux Ubuntu 18.10
Reference Device	Lepu Medical PalmECG PC-80D (Three-lead ECG Monitor)
Number of Antennas	One transmitter, two receivers
Default Transmission Power	24 dbm
Carrier Frequency	5.8 GHz
Transmission Rate	1000 packets/s

## Data Availability

Data are contained within the article.

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
