# Peer review of "Non-Contact Heart Rate Monitoring Method Based on Wi-Fi CSI Signal"

_sensors, 2024, doi:10.3390/s24072111_

Round 1

Reviewer 1 Report

Comments and Suggestions for Authors

Comments on the Quality of English Language

Minor editing of English language required throughout the entire paper.

Reviewer 2 Report

Comments and Suggestions for Authors

The authors have performed excellent work in this paper. RF-based solutions are state-of-the-art, especially in healthcare applications. I have a few concerns about the paper which I have mentioned in the comments below. Please address them in detail.

1. In the manuscript, please mention the exact WiFi devices and antennas used for the experiment. Upload real pictures of the devices if possible so it helps the reader to identify them. Also make a table which mentions all the hardware used for the experiments, their exact models and the properties they can work on such as frequency etc.

2. A figure shall be added in the paper which shows the clear samples of the final heart rate estimations. If possible, use a heart rate estimation belt or any other device as a ground truth, which can clarify if the signal estimated is correct.

3. Humans are involved in this experiment. First, was the ethical approval obtained to experiment on human data? Please mention ethical approval at the end of the manuscript. Also, make a table which shows the number of human participants, their body weight/mass, age and gender so that the reader can be clarified. Also, each participant shows a sample of heart rate estimation so the differences between each participant can be shown.

4. Below I have suggested some relevant papers based on RF sensing, discuss them in your related work section.

Saeed, U., Shah, S. Y., Zahid, A., Ahmad, J., Imran, M. A., Abbasi, Q. H., & Shah, S. A. (2021). Wireless channel modelling for identifying six types of respiratory patterns with sdr sensing and deep multilayer perceptron. IEEE Sensors Journal21(18), 20833-20840.

Rehman, M., Shah, R. A., Ali, N. A. A., Khan, M. B., Shah, S. A., Alomainy, A., ... & Abbasi, Q. H. (2023). Enhancing System Performance through Objective Feature Scoring of Multiple Persons’ Breathing Using Non-Contact RF Approach. Sensors23(3), 1251.

Comments on the Quality of English Language

English is okay, overall! Minor errors. Use tools like Grammarly to eliminate grammatical issues.

Reviewer 3 Report

Comments and Suggestions for Authors

The manuscript is supposed to capture a heart pulse correlated signal by Wi Fi RF carriers amplitude and phase variations. The principle is not original at all and the only declared novelty should be the use of the amplitude and phase of some subcarriers, selected based on their signal to noise ratio.

The main problems is that no comparison with one of the many non invasive pulse related signals is provided. Was this done, the authors probably would have noticed the improbable beat correlation or the signal they derived after a number of cumbersome passages. This is mandatory ahead of moving to heart rate errors, since heart rate definition is not univocal and is based on how long or the count of beats is done.

Comments on the Quality of English Language

The editing was not bad, at least.
